# Advances in Understanding the Impact of Human Gut Microbiota on Chemotherapy-Induced Neutropenia

**DOI:** 10.3390/biomedicines14010055

**Published:** 2025-12-26

**Authors:** Mengyuan He, Liangkang Lin, Cheng Ouyang, Su Liu, Chun Chen

**Affiliations:** Pediatric Hematology Laboratory, Division of Hematology/Oncology, Department of Pediatrics, The Seventh Affiliated Hospital of Sun Yat-sen University, Shenzhen 518107, China; hemengyuan@sysush.com (M.H.); linliangkang@sysush.com (L.L.);

**Keywords:** neutrophils, chemotherapy-induced neutropenia, gut microbiota

## Abstract

Neutrophils play a crucial role in defending against bacterial and fungal pathogens; however, chemotherapy reduces neutrophil counts, increasing infection risk and worsening cancer treatment outcomes. Emerging evidence highlights the gut microbiota as a promising biomarker and therapeutic target for chemotherapy-induced neutropenia (CIN). Here, we review current knowledge regarding the relationship between the gut microbiota and CIN, summarizing the mechanisms by which the microbiota influence neutrophil dynamics, current therapeutic approaches, and limitations in regard to preserving microbiota stability. This review offers a theoretical foundation for “gut-protective” chemotherapy, potentially facilitating personalized treatments, although clinical translation remains challenging.

## 1. Introduction

Cancer is a critical factor affecting the human lifespan. It is a leading global cause of death and incurs substantial societal and macroeconomic costs. Globally, there were close to 18.5 million new cases of cancer and 10.4 million deaths caused by cancer in 2023, indicating that approximately 35 people were diagnosed with cancer every minute [1]. Despite decades of progress in innovative anticancer therapies, chemotherapy remains the primary clinical treatment for cancer [2]. However, chemotherapy is non-specific, destroying not only cancer cells but also healthy cells, leading to chemotherapy-induced neutropenia (CIN) and, potentially, prolonged delays in neutrophil recovery (DNR), particularly in the context of hematologic malignancies such as acute lymphoblastic leukemia (ALL) and acute myeloid leukemia (AML) [3,4,5].

Evidence supporting the association between the gut microbiota and chemotherapy outcomes is accumulating. A systematic review including 22 studies found that changes in the abundance of distinctive bacterial taxa of the gut microbiota—such as relatively higher abundances of *Streptococcus mutans*, *Enterococcus casseliflavus*, and *Bacteroides* in lung tumors—might be associated with favorable clinical responses to chemotherapy [6]. A recent review also discussed the role of gut microbiota and metabolites in cancer chemotherapy and highlighted that targeting gut microbiota to enhance the efficacy of chemotherapy and reduce its toxicity might be a promising tumor treatment strategy [7]. Furthermore, Yang et al. summarized the correlation between gut-microbiota-derived metabolites and cancer from the perspectives of cancer progression and therapy [8]. They suggested that the application of microbial metabolites in radiotherapy, chemotherapy, and immunotherapy could significantly enhance the effects of cancer treatment, with the major benefits primarily including overcoming drug resistance, ameliorating treatments’ toxic side effects, and activating the immune system in patients with cancer [8]. Other researchers have conducted similar reviews [9,10,11]. However, few focused on the relationship between gut microbiota and CIN.

Neutrophils, abundant innate immune cells, rapidly respond to infection, injury, and inflammation, playing a pivotal role in host defense against bacterial and fungal pathogens [12,13]. CIN and DNR increase infection risk and delay chemotherapy cycles, significantly worsening the prognoses of patients with ALL or AML. Recent studies have demonstrated that gut microbiota dysbiosis persists throughout leukemia treatment and correlates with increased mortality [14,15]. Additionally, Sardzikova et al. analyzed the gut microbiota composition of 18 pediatric oncology patients undergoing hematopoietic stem cell transplantation (HSCT) and found that significant decreases in alpha diversity were associated with febrile neutropenia [16]. Sørum et al. investigated the interaction between gut microbiota and neutrophil dynamics in 51 children recently diagnosed with ALL, finding that reduced abundance of specific intestinal commensals and *Enterococcus* overgrowth were correlated with delayed neutrophil reconstitution and increased chemokine signaling, suggesting microbiota disruption contributes to prolonged CIN [15,17]. However, the precise mechanisms underlying microbiota–neutrophil interactions remain incompletely understood.

In this review, we summarize and discuss the adverse effects of CIN and DNR induced by chemotherapeutic agents. Chemotherapy can cause gut microbiota dysbiosis, which may be closely associated with DNR. We further explore the key mechanisms connecting gut microbiota dysbiosis with CIN or DNR. Finally, currently available strategies for alleviating gut microbiota dysbiosis are comprehensively summarized and discussed. In this study, we provide a foundation for future research investigating the mechanisms underlying the relationship between gut microbiota dysbiosis and CIN or DNR, as well as the clinical significance of these mechanisms for developing and optimizing microbiota-sparing interventions, ultimately aiming to enhance chemotherapy outcomes, particularly for patients with ALL or AML.

## 2. Chemotherapy and Chemotherapy-Induced Neutropenia

Chemotherapy, the primary treatment modality for cancer, can suppress bone marrow function, thereby reducing neutrophil counts and causing prolonged DNR. DNR significantly increases infection risk. Galloway-Peña et al. reported that patients with AML experiencing DNR after chemotherapy were approximately four times more likely to develop infections (HR [hazard ratio] = 4.55, 95%CI [confidence interval] = 1.73–11.93) [18]. Additionally, DNR following CAR-T cell therapy is associated with an elevated risk of infection and significantly different one-year overall survival rates between patients with CIN for ≤38 days (85%) and those with CIN for >38 days (44%, *p* = 0.029) [19].

Numerous case reports indicate that severe and uncommon infections may follow CIN. Koizumi et al. reported the case of a 54-year-old Japanese woman with AML who developed Bacillus cereus bacteremia and meningitis accompanied by severe leukocytopenia during consolidation chemotherapy [20]. The patient’s consciousness rapidly deteriorated, necessitating transfer to the intensive care unit [20]. Similarly, Zhang et al. reported the case of a 46-year-old Chinese male patient with rectal cancer who developed febrile neutropenia followed by an acute hepatitis E infection concurrent with liver and lung metastases [21]. Another case study involving a 58-year-old female patient with glioblastoma showed that CIN could lead to sepsis caused by phlegmonous gastritis [22].

Numerous studies evaluating the association between CIN and infection are summarized in Table 1.

**Table 1 biomedicines-14-00055-t001:** Association between chemotherapy-induced neutropenia and infections.

Study	Study Design	Country	Patients	Outcomes
Gomez et al. [23]	Single-center, prospective observational study	Spain	157 patients with hematologic cancers or solid tumors	In total, 45.0% developed severe neutropenia. A total of 1.7% met sepsis criteria, with all cases progressing to septic shock.
Frairia, et al. [24]	Retrospective cohort study	Italy	334 adult AML patients receiving intensive induction chemotherapy	Overall, 19% had bloodstream infections, 10% had pneumonia, 3% had neutropenic enterocolitis, and 1.4% had central venous catheter infections. In total, 5% developed SIRS, with 24% progressing to sepsis and 51% progressing to septic shock.
Fedhila et al. [25]	Prospective, longitudinal descriptive study	France	32 pediatric patients with hematological or solid tumors presenting with neutropenic fever	Overall, 18% had microbiologically documented fever; 2% died from septic shock.
de Jonge et al. [26]	Non-inferiority, open-label, multicenter, randomized trial	Netherlands	281 adults receiving intensive chemotherapy or HSCT for hematological malignancies with fever of unknown origin during high-risk CIN	17% treatment failure was observed (which was defined as the occurrence of recurrent fever or carbapenem-sensitive infection on days 4–9, as well as the development of septic shock, respiratory failure, or death prior to neutrophil recovery)
de la Court et al. [27]	Retrospective cohort study	Netherlands	372 adults receiving chemotherapy for hematological malignancies with protracted CIN	Bloodstream infections occurred in 20.1% of patients.
Charakopoulos et al. [28]	Case report	Greece	46-year-old male with ALL (t(12;17) (p13;q21)/TAF15-ZNF384)	Febrile neutropenia on day 22 of chemotherapy, followed by fulminant Aeromonas hydrophila soft-tissue infection, was reported.

CIN, chemotherapy-induced neutropenia. AML, acute myeloid leukemia. SIRS, systemic inflammatory response syndrome. HSCT, hematopoietic stem-cell transplantation.

DNR also leads to delayed treatment of cancer patients. A study involving 51 children recently diagnosed with ALL highlighted how prolonged CIN was associated with delayed treatment cycles [17]. Such delays not only prolong treatment but also potentially reduce therapeutic efficacy. A multicenter, randomized trial reported death within 30 days after neutrophil recovery (including infection-related death for five patients), underscoring the serious potential consequences of DNR [26]. Lackraj et al. reported that clonal hematopoiesis was associated with longer neutrophil recovery times, correlating with inferior 5-year overall survival rates after the first relapse (39.4% vs. 45.8%, *p* = 0.043) [29]. Similarly, a multicenter retrospective study noted increased 1-year non-relapse mortality among aplastic patients (with continuous severe neutropenia for ≥14 days), with poor progression-free survival and overall survival rates (1-year PFS [progression-free survival]: 26%; 1-year OS [overall survival]: 46%) [30].

Collectively, chemotherapy-induced DNR and CIN increase infection risks and prolong chemotherapy cycles, significantly impacting patient prognosis. Notably, management of febrile neutropenia in cancer patients remains challenging for emergency departments struggling to comply with international guidelines [31].

## 3. Chemotherapeutic Agents and Microbiome Signatures

Cancer chemotherapy hinges on a panel of structurally and functionally distinct agents, each endowed with a unique mechanism of action to selectively target malignant cells. Alkylating agents (such as cyclophosphamide), antimetabolites (5-fluorouracil, methotrexate), anthracyclines, platinum compounds, and taxanes are the most commonly used categories of antineoplastic agents in clinical practice. Nevertheless, these therapeutic regimens induce profound and often deleterious perturbations in the gut microbiome. Herein, we present a detailed overview of the clinically relevant associations between chemotherapeutic agents and their corresponding gut microbiome signatures.

Zeng et al. demonstrated that cyclophosphamide (Cy) administration elicited a marked shift in gut microbial composition at the phylum level: the relative abundance of Firmicutes decreased significantly from 58.84% to 47.97%, whereas that of *Bacteroidetes*, *Actinobacteria*, *Verrucomicrobia*, and *Proteobacteria* was concurrently elevated [32]. At the genus level, this chemotherapeutic intervention led to a depletion of beneficial taxa, including *Lactobacillus*, with 12 genera (e.g., *Akkermansia* and *Ruminococcus*) identified as distinct microbial biomarkers of Cy exposure [32]. Moreover, Cy treatment disrupted the intrinsic inter-microbial interaction network, as evidenced by a pronounced negative correlation between *Lactobacillus* and *Akkermansia*, ultimately culminating in a state of gut dysbiosis [32]. Corroborating these findings, Huang et al. reported that Cy administration profoundly perturbed gut microbiota homeostasis in murine models, as characterized by a dual microbial shift: a reduction in the abundance of beneficial phyla (*Bacteroidetes* and *Verrucomicrobia*) and genera (*Bacteroides* and *Oscillospira*) coupled with an expansion in the quantity of potentially pathogenic taxa (*Proteobacteria*) and the aberrant overgrowth of *Lactobacillus* [33]. Beyond compositional alterations, Cy treatment exerted deleterious effects on microbial community structure: it markedly impaired both alpha diversity (manifested as reduced species richness and community diversity) and beta diversity (indicating compromised overall structural stability of the gut microbiome) [33]. Concomitantly, Cy suppressed the biosynthesis of short-chain fatty acids (SCFAs)—key microbial metabolites encompassing acetate, propionate, and isovalerate—and induced a substantial increase in the number of differentially abundant bacterial taxa [33]. Notably, consistent observations have been documented in numerous independent studies, further validating the disruptive effects of Cy on gut microbial homeostasis and diversity [34,35,36].

5-fluorouracil (5-FU) has also been demonstrated to exert significant multifaceted regulatory effects on the gut microbiota. Specifically, Trepka and colleagues reported that fluoropyrimidine-based chemotherapy for colorectal cancer substantially reshaped gut microbial community structure: this remodeling was characterized by a significant reduction in microbial α-diversity, alongside a specific enrichment in bacterial taxa harboring the preTA operon. Notably, the relative abundance of these preTA operon-containing bacteria was found to be negatively correlated with the severity of fluoropyrimidine-induced chemotherapy-related toxicity [37]. Furthermore, supplementation with bacteria engineered to express the preTA operon was shown to mitigate 5-FU-associated drug toxicity, thereby identifying a novel microbial target with potential utility for both predicting and performing targeted interventions for chemotherapy-induced adverse effects [37]. In a complementary murine study, Menezes-Garcia et al. determined that 5-FU administration similarly induced a profound shift in gut microbiota composition, resulting in a marked increase in the relative abundance of *Enterobacteriaceae* (including *E. coli*) in both fecal samples and small-intestine tissues [38]. This microbial dysbiosis was concomitantly associated with overt intestinal tissue damage [38]. Critically, prophylactic treatment of mice with ciprofloxacin—an antibiotic known to suppress *Enterobacteriaceae* outgrowth—led to a significant attenuation of 5-FU-induced tissue injury. This finding strongly indicates that increasing the quantity of *Enterobacteriaceae* is a necessary prerequisite for the development of 5-FU-mediated inflammatory responses in the gastrointestinal tract [38]. Consistent with these observations, Yang Q et al. further confirmed that 5-FU treatment induced significant alterations in both the α-diversity and overall community composition of the gut microbiota [39]. At the genus level, comparative analysis against control groups revealed a significant increase in the relative abundance of *Lachnospiracea*_NK4A136, *Bacteroides*, *Odoribacter*, *Mucispirillum*, and *Blautia* following 5-FU exposure [39]. Collectively, these findings are congruent with a growing body of recent investigations that consistently highlight the substantial impact of gut microbial homeostasis on 5-FU and underscore the potential of targeting the gut microbiota to modulate 5-FU-associated toxicity [40,41].

Methotrexate (MTX), which can modulate the human gut microbiota, is closely linked to suboptimal treatment outcomes and an elevated risk of adverse toxic effects. Nayak et al. found that MTX inhibited the growth of 45 isolated gut bacterial strains, with the *phylum Bacteroidetes* displaying heightened sensitivity in vitro [42]. Additionally, the high-dose MTX group exhibited a marked reduction in *Bacteroidetes* abundance, an effect not significantly impacted by concurrent folic acid supplementation, administration routes, or variations in donor microbiota [42]. Moreover, a pre-treatment analysis of gut microbiota revealed significant compositional discrepancies between MTX non-responders and responders. Specifically, MTX non-responders exhibited a significantly higher abundance of *Firmicutes*, a lower abundance of *Bacteroidetes*, and consequently an elevated *Firmicutes/Bacteroidetes* ratio compared to responders (*p* < 0.05) [43]. Additionally, the gut microbiota of non-responders was characterized by increased enrichment of *Euryarchaeota phylum*, unclassified *Clostridiales/Clostridiales incertae sedis* XIII, and *Escherichia/Shigella* (*p* < 0.05) [43]. Furthermore, the intestinal microbiota of mice showed a time-dependent decrease in diversity and marked compositional alteration after MTX treatment. Among all microbial taxa, *Bacteroidales* displayed the most striking changes, with a significant reduction in the abundance of specific *Bacteroides* species (e.g., *Bacteroides fragilis*) [44]. Similar microbiota-associated patterns have also been identified with regard to other chemotherapeutic agents, including anthracyclines [45,46], platinum compounds [47,48,49], and taxanes [50,51].

## 4. Gut Microbiota Dysbiosis and Chemotherapy-Induced Neutropenia

Numerous studies have demonstrated that chemotherapy induces gut microbiota dysbiosis by disrupting the structure and function of the intestinal microbiome, which consequently features significantly reduced microbial diversity and altered abundance of specific bacterial taxa [52,53,54]. Moreover, the intestinal microbiome influences hematopoiesis and the treatment outcomes of hematologic diseases [55]. Thus, gut microbiota dysbiosis may be closely associated with DNR, particularly among patients undergoing immunosuppressive therapy or chemotherapy.

There is a growing body of evidence revealing specific microbial signatures associated with the risk of febrile neutropenia (FN) in patients undergoing chemotherapy or HSCT. The abundance of beneficial bacteria, such as *Bacteroidetes* and *Faecalibacterium*, was significantly reduced after chemotherapy, while the abundance of potential pathogenic bacteria, including *Clostridiaceae*, *Streptococcaceae*, and *Enterococcaceae*, significantly increased [56]. A baseline relative abundance of *Proteobacteria* ≥ 0.01% was identified as an independent predictor of subsequent FN (HR = 2.12; incidence rate = 67%) [56]. Composition Type 3, which predominantly consists of *Enterococcaceae*, *Streptococcaceae*, or *Lactobacillaceae*, was confirmed as a significant risk factor for FN, diarrheal illness, and all types of infections [56]. Furthermore, when *Enterococcaceae* accounted for ≥30%, the risks of subsequent FN (HR = 2.97) and diarrheal illness (HR = 4.23) were significantly elevated [56]. Additionally, an analysis of 119 patients with neutropenia following HSCT revealed that 53% (63 patients) developed subsequent fevers [57]. The fecal microbiomes of these febrile patients were characterized by a significant increase in the abundance of mucin-degrading bacteria (*Akkermansia muciniphila* and *Bacteroides* genus) along with enhanced mucin glycan degradation capacity [57]. Moreover, febrile patients exhibited distinct changes in microbiome composition before and after HSCT, namely, an increase in the quantity of *Akkermansia muciniphila* and a decrease in *Bacilli* numbers [57]. In contrast, afebrile patients showed higher abundances of *Bacilli* and *Erysipelotrichales*, with no significant alterations in the relevant bacterial taxa between the pre- and post-HSCT periods [57]. Furthermore, Rashidi et al. found that 1 day pre-NF onset, their NF group had significantly lower intestinal *Blautia* quantities but higher *Bifidobacterium* and *Veillonella* abundances than the control group [58]. Furthermore, *Blautia*’s NF-protective effect presented strong temporal specificity: it was only significant when there was a ≤24 h interval between stool sampling and NF onset [58].

Studies have investigated the associations between the gut microbiome and chemotherapy, covering three core dimensions: resistome profiles, infection risk, and microbiome-associated correlates of chemotherapy response. Firstly, 16S rRNA gene sequencing of 158 patient samples revealed an increase in the abundance of facultative anaerobic bacteria such as *Escherichia coli* in patients with diffuse large B-cell lymphoma [59]. These bacteria are associated with opportunistic pathogenic functions, including antibiotic resistance. Notably, high abundance of *Enterobacteriaceae* correlates with significantly shortened PFS and an increased risk of relapse or progression and serves as an independent risk factor for PFS [59]. Beyond prognostic value, the gut microbiome also modulates chemotherapy-associated adverse events. A Vietnam Breast Cancer Study showed that elevated gut microbial α-diversity prior to chemotherapy was significantly associated with a lower incidence of severe hematological toxicity in patients undergoing neoadjuvant or adjuvant chemotherapy [60]. Taxa-level analysis further refined this association: taxa within *Synergistota* were linked to a reduced risk of severe neutropenia, while taxa belonging to *Firmicutes C* and *Firmicutes I* correlated with an increased risk of this adverse event [60]. In addition to modulating prognosis and adverse reactions, the gut microbiome can also regulate the therapeutic efficacy of chemotherapy drugs. Yang et al. indicated that, compared to gut-only probiotic intervention, “intestinal-vaginal” probiotic administration significantly enhanced the anticancer efficacy of 5-fluorouracil by upregulating the p53 pathway [39]. This regulatory effect on chemotherapy efficacy is further supported by mechanistic evidence from animal models. Tumor-bearing mice that were germ-free or had been treated with antibiotics to kill Gram-positive bacteria exhibited a reduction in pathogenic T helper 17 (pT(H)17) responses, and their tumors were resistant to Cy [61]. Adoptive transfer of pT(H)17 cells partially restored the antitumor efficacy of Cy [61], suggesting that the gut microbiota helped shape the anticancer immune response.

There is an accumulating amount of evidence supporting the notion that the gut microbiome contributes significantly to neutrophil regeneration following chemotherapy [53,62]. Based on patient cohort studies, Sørum et al. proposed that DNR in ALL treatment is directly related to gut microbiota disruption [17]. The proliferation of opportunistic bacteria, such as *Enterococcus* species, and the depletion of certain intestinal commensals were identified as risk factors, suggesting that an imbalance in microbiota composition influences neutrophil dynamics [17]. In addition, microbial diversity was significantly reduced in AML patients during chemotherapy and continued to decline post-transplantation, a trend linked to increased mortality [14,63]. Similarly, antibiotic-induced dysbiosis accelerated AML progression and indirectly modulated sensitivity to chemotherapeutic agents, effects that are potentially reversible via fecal microbiota transplantation (FMT) [64,65]. Rashidi et al. reported that circulating metabolomic profiles after NF episodes in AML patients exhibited a minimal signature of 18 metabolites, 13 of which were microbiota-derived, including markers of intestinal epithelial health and gut-protective dietary tryptophan metabolites [66]. The quantity of these metabolites declined concurrently with biologically consistent reductions in *mucolytic* and *butyrogenic bacterial* abundance, indicating that post-NF metabolomic shifts primarily involve the loss of microbiota-derived protective metabolites rather than an increase in detrimental metabolites [66].

A comprehensive understanding of the mechanisms linking gut microbiota dysbiosis to DNR is required in order to optimize chemotherapy regimens and develop preventive and therapeutic strategies. The principal mechanisms by which gut microbiota influence CIN are discussed in this section.

### 4.1. Immune-Regulatory Pathways

There is an increasing amount of evidence demonstrating intricate crosstalk between the host immune system and gut microbiota during neutrophil recovery. Zhang et al. showed that highly efficient depletion and dramatic alterations in gut microbiota composition led to significant and selective reductions in neutrophil numbers in both the circulation and bone marrow [67]. Compared to specific pathogen-free animals, germ-free mice exhibited widespread changes in innate and adaptive immune cell populations, strongly indicating that neutrophil aging is regulated by the gut microbiome [67]. Employing mouse models of 5-FU-induced neutropenia and syngeneic hematopoietic stem cell transplantation, Chen et al. demonstrated that neutrophil recovery was significantly delayed in IL-17A-deficient or T-cell-deficient RAG1-/- mice [68]. Additionally, decontamination of the gut with oral antibiotics reduced neutrophil recovery and suppressed IL-17A production by T cells [68], suggesting that the microbiota promotes granulocyte production via the Th17-IL-17A axis. Furthermore, recent studies indicate that the gut microbiota may regulate neutrophil homeostasis through control of reactive granulopoiesis and immunological recovery [17,68]. This interaction is part of a complex dialog between the gut–immune axis and neutrophils, where neutrophils regulate microbiota composition through inflammatory pathways, while dysbiosis impairs neutrophil recruitment and maturation [69,70].

### 4.2. Metabolic Pathways

Gut microbiota dysbiosis influences neutrophil function and recovery through several metabolic pathways.

SCFAs (primarily acetate, propionate, and butyrate), produced by microbial fermentation of dietary fibers and polysaccharides, play critical roles in intestinal epithelial health, host metabolism, and immune modulation [71]. Clinical trials have demonstrated that fecal SCFA levels decline following chemotherapy treatment [72,73]. Dysbiosis may cause reductions in levels of SCFAs such as butyrate, a metabolite crucial for immune regulation [74]. In animal models, Li et al. demonstrated that butyrate suppresses the neutrophil production of pro-inflammatory mediators via its pan-histone deacetylase inhibitory activity [75]. Zucoloto et al. showed that mice colonized with Escherichia coli ASF360, a high-lactate-producing strain, exhibited significantly improved recruitment of neutrophils to the liver [76]. Xuan et al., using 6–8-week-old male C57BL/6 mice, explored how fatty-acid-binding protein 4 (FABP4), influenced by gut-microbiota-derived SCFAs, delayed neutrophil apoptosis through endoplasmic reticulum stress, thereby potentiating inflammatory factors and causing lung epithelial cell damage [77]. A multicenter randomized study further identified significant correlations between post-chemotherapy diarrhea severity, FN incidence, increased *Clostridioides difficile* abundance, and reduced fecal acetic acid concentrations [78]. Applying amplicon sequencing, viral metagenomics, and targeted metabolomics to stool samples from 78 stem cell transplant patients, Thiele Orberg et al. established an immunomodulatory metabolite risk index associated with improved survival and lower relapse risk, observing elevated activity of SCFA biosynthesis pathways (particularly butyrate production via BCoAT[butyryl-coenzyme A:acetate coenzyme A-transferase]) in low-risk patients [79].

Additionally, disruptions in the urea cycle and pyrimidine metabolism cannot be overlooked. Fernandez-Sanchez et al., using clinical data and the Quade test, observed significant depletion of 21 specific stool metabolites, including bacterial cell wall component *N*-acetylglucosamine, urea-cycle intermediates citrulline and ornithine, and the pyrimidine metabolite 2′-deoxyuridine, in neutropenic subjects relative to controls [80]. Neutrophil regeneration may be delayed by the absence of these metabolites due to their essential roles as substrates in neutrophil proliferation and DNA synthesis.

### 4.3. Barrier Protection Function

The barrier-protective function of intestinal mucosa is another critical factor. There is an accumulating amount of evidence showing that chemotherapeutic agents damage intestinal mucosa, causing gut microbiota dysbiosis. Wong et al., using a mouse model given consecutive daily irinotecan injections, found that genetic deletion of TLR4 increased TLR9 expression, exacerbating intestinal mucositis severity and late-onset diarrhea [81]. In mice, Zhou et al. demonstrated that methotrexate treatment caused significant, time-dependent alterations in gut microbiota diversity, composition, and abundance, with *Bacteroidales* exhibiting the greatest shifts, correlated proportionally with increased intestinal macrophage density [44]. Additionally, studies using mouse models have demonstrated that intraluminal neutrophils protect epithelial integrity by limiting pathogen invasion of intestinal epithelial cells [82]. Gut microbiota dysbiosis increases intestinal permeability; facilitates pathogen translocation, including potential lung colonization; activates systemic inflammatory signals (e.g., the “gut–lung axis”); depletes neutrophil reserves; and delays neutrophil recovery [52,83,84].

Chemotherapeutic agents (such as irinotecan, fluorouracil, and cyclophosphamide) trigger a vicious cycle of “neutropenia–intestinal mucosal barrier injury–microbiome dysbiosis–diarrhea/mucositis” while eliminating tumor cells. Chemotherapy directly damages intestinal epithelial cells and bone marrow hematopoietic function initially, resulting in shortened intestinal villi, decreased expression of tight junction proteins, and a sharp reduction in neutrophil numbers, thus impairing the innate immune defense of the intestine [11,17]. This process disrupts the intestinal microenvironment, leading to reduced abundance of beneficial bacteria (such as *Bifidobacterium* and certain strains of *Lactobacillus*) and excessive proliferation of pro-inflammatory bacteria and functionally abnormal bacteria (β-glucuronidase-expressing *Lactobacillus reuteri*, *Escherichia coli-Shigella*, etc.) [85,86]. Moreover, β-glucuronidase-positive bacteria bioactivate the non-toxic SN38G, turning it into the cytotoxic metabolite SN38, to exacerbate mucosal erosion, while the depletion of beneficial bacteria reduces the production of protective SCFA and thus impairs intestinal stem cell regeneration and neutrophil function [85,87]. Furthermore, intestinal mucosal barrier injury further promotes bacterial translocation and the proliferation of aerobic pathogenic bacteria. Inflammatory factors suppress immune and repair functions, and neutropenia cannot control the growth of pathogenic bacteria [88].

Through interference with neutrophil recruitment, activation, and function via various mechanisms, such as alterations in metabolic products, immune signaling, and impaired intestinal barrier function, gut microbiota dysbiosis ultimately contributes to DNR and/or functional impairments. This information on these pathways provides theoretical support for the development of microbiota-targeted therapeutic strategies, such as probiotics, FMT, and dietary interventions.

## 5. Therapeutics and Limitations

Chemotherapy significantly disrupts the gut microbiota [78]. The resulting disruption during cancer treatment markedly elevates the risk of secondary complications. Although experimental data indicate the increasing clinical potential for microbiota restoration [62,89,90,91,92], oncologists remain cautious about clinical application due to potential infectious complications. Currently, probiotics, postbiotics, synbiotics, and FMT are the predominant strategies employed for gut microbiota modulation.

### 5.1. Probiotics, Postbiotics, and Synbiotics

Probiotics are live microorganisms that, when administered in adequate amounts, confer health benefits to the host [93], whereas postbiotics are described as “preparations of inanimate microorganisms and/or their components that confer a health benefit on the host.” [94]. The International Scientific Association for Probiotics and Prebiotics (ISAPP) recently updated the definition of synbiotics, changing it to “a mixture comprising live microorganisms and substrate(s) selectively utilized by host microorganisms that confers a health benefit on the host” [95]. Recently, considerable research has been directed towards maintaining gut microbiota balance via supplementation with probiotics, prebiotics, or synbiotics to reduce chemotherapy-related adverse effects. The primary studies investigating the use of probiotics, prebiotics, or synbiotics in cancer patients are summarized in Table 2.

Probiotics and postbiotics (heat-killed and live *Limosilactobacillus reuteri* PSC102) have been demonstrated to have multiple beneficial effects. In Cy-induced immunosuppressed Sprague-Dawley rats, these agents significantly increased counts of white blood cells, granulocytes, lymphocytes, and neutrophils; enhanced neutrophil migration and phagocytosis, splenocyte proliferation, and levels of T lymphocyte subsets (CD4+, CD8+, CD45RA+, and CD28+); upregulated immune factors (IL-2, IL-4, IL-6, IL-10, IL-12A, TNF-α, and IFN-γ); and modulated gut microbiota composition by increasing the relative abundances of *Bacteroidetes and Firmicutes* while decreasing Proteobacteria levels [96]. Additionally, the probiotic *Lactobacillus delbrueckii* CIDCA 133 strain reduced infiltration of neutrophils into small intestinal mucosa and ameliorated intestinal epithelial damage induced by 5-FU, effects that were associated with downregulated inflammatory markers (Tlr2, Nfkb1, Il12, Il17a, Il1b, and Tnf) and upregulated immunoregulatory cytokine (Il10) and epithelial barrier markers (Ocln, Cldn1, Cldn2, Cldn5, Hp, and Muc2) [97]. However, several studies have reported inconsistent results. Gorshein et al. [98] conducted the first randomized trial of probiotic (*Lactobacillus rhamnosus* GG) supplementation in patients undergoing HSCT and found no significant probiotic-induced changes in the gut microbiome or incidences of acute graft-versus-host disease (aGVHD). Mehta et al. reported a cautionary case involving *Lactobacillus acidophilus*-induced sepsis in a mantle-cell lymphoma patient undergoing HSCT, highlighting the potential risks associated with probiotic use in immunocompromised patients [99].

Compared to probiotics, synbiotics appear more effective at altering microbiota composition, minimizing bacteremia risks in severely immunocompromised patients, and enhancing HSCT outcomes. Fukaya et al. conducted a randomized controlled trial demonstrating that synbiotic-mediated suppression of bacterial translocation was partly attributed to increased fecal acetic acid concentrations, which may improve intestinal barrier function, alongside reduced fecal pH [100]. Yazdandoust et al. also indicated that synbiotic supplementation (a mixture containing high concentrations of seven safe bacterial strains and fructo-oligosaccharides) before and during conditioning in allo-HSCT patients could reduce the incidence and severity of aGVHD by inducing CD4+ CD25+ Foxp3+ regulatory T cells, thereby improving transplantation outcomes [101]. An ancillary study derived from a randomized controlled trial involving 73 patients with esophageal cancer receiving prophylactic antibiotics or synbiotics (*Lacticaseibacillus paracasei strain Shirota*, *Bifidobacterium breve strain Yakult*, and galacto-oligosaccharides [LBG]) combined with enteral nutrition (LBG+EN) found higher abundances of *Anaerostipes hadrus and Bifidobacterium pseudocatenulatum* associated with reduced severity or incidence of neoadjuvant chemotherapy (NAC)-related adverse events (e.g., absence of febrile neutropenia and reduced diarrhea). Specifically, pre-NAC A. hadrus abundance in the LBG+EN subgroup correlated with a lower FN risk, and post-NAC A. hadrus abundance positively correlated with intestinal concentrations of acetic and butyric acid [102].

**Table 2 biomedicines-14-00055-t002:** Use of probiotics, postbiotics, or synbiotics in cancer patients.

Study	Study Design	Intervention	Cancer Type	Subjects	Outcomes
Fukaya et al. [100]	RCT (*n* = 42)	Synbiotics	Esophageal Cancer	Human	Reduced incidence of grade 3 gastrointestinal toxicity and bacteremia during chemotherapy.
Motoori et al. [78]	RCT (*n* = 81)	Synbiotics combined with enteral nutrition	Esophageal Cancer	Human	Significantly reduced incidence of grade 4 neutropenia. Increased alpha diversity and acetic acid concentration compared to antibiotic group.
Motoori et al. [103]	RCT (*n* = 61)	Synbiotics	Esophageal Cancer	Human	Lower incidence of febrile neutropenia (synbiotics 10/30 vs. control 19/31, *p* = 0.029).
Stene et al. [104]	Controlled trial (*n* = 30)	Synbiotics	Colorectal Cancer	Human	Greater preservation of bacterial species (25.1% reduction vs. 55.4% in prebiotic group).
Eghbali et al. [105]	RCT (*n* = 113)	5 × 10^9^ CFU LactoCare synbiotic	ALL	Human	Lower incidence of chemotherapy-induced diarrhea relative to the placebo group (3.7%–0% vs. <10%–13.5%, days 1–7).
Yazdandoust et al. [101]	RCT (*n* = 40)	Synbiotic mixture (7 bacterial strains plus fructo-oligosaccharides)	allo-HSCT	Human	Lower incidence of severe aGVHD (0% vs. 25% control). Improved 12-month overall survival (90% vs. 75%).
Mizutani et al. [106]	RCT (*n* = 12)	Synbiotic (*Bifidobacterium longum* BB536 and guar gum)	allo-HSCT	Human	Reduced duration of grade ≥ 3 diarrhea (2.5 vs. 6.5 days) and hospital stay (31.5 vs. 43 days). No synbiotic-related infections observed.
Batista et al. [97]	NP	Probiotics (heat-killed *L.delbrueckii* CIDCA 133)	5-Fluorouracil drug-induced Mucositis	Mice	Reduced infiltration of neutrophils into intestinal mucosa. Ameliorated intestinal epithelial damage caused by 5-FU.
Nobre et al. [107]	NP	Probiotics (*paraprobiotic E. faecalis formulation*)	Irinotecan-induced intestinal mucositis	Mice	Inhibited irinotecan-induced translocation of bacteria into blood.
Ali et al. [96]	NP	Probiotics and postbiotics (Limosilactobacillus reuteri PSC102)	Cyclophosphamide- treated rats	Mice	Improved absolute neutrophil counts, enhanced neutrophil migration/phagocytosis, and modulated microbiota (increased abundance of Bacteroidetes/Firmicutes and decreased abundance of Proteobacteria).
Tang, et al. [108]	NP	Probiotics (*Lactobacillus reuteri* and *Clostridium butyricum* Miyairi 588)	5FU-induced colitis diarrhea	Rats	Reduced neutrophil infiltration and inflammation; preserved mucosal barrier integrity through antioxidant and anti-apoptotic effects; and modulated cytokine and aquaporin expression.
Gorshein et al. [98]	RCT (*n* = 31)	Probiotics (*Lactobacillus rhamnosus* GG)	allo-HSCT	Human	No significant alteration in gut microbiome or protection against GVHD after allo-HSCT.
Mehta et al. [99]	Case report	Probiotics (*Lactobacillus acidophilus*)	Autologous HSCT	Human	Developed *Lactobacillus acidophilus*-induced sepsis.
Fukushima et al. [109]	Retrospective study (*n* = 40)	Clostridium butyricum MIYAIRI 588	allo-HSCT	Human	Contributed to maintenance of gut microbiota diversity early after HSCT.

aGVHD, acute graft-versus-host disease; ALL, acute lymphoblastic leukemia; allo-HSCT, allogeneic hematopoietic stem cell transplantation; NP, no report; RCT, randomized controlled trial.

### 5.2. FMT

Numerous studies have demonstrated that the intestinal microbiota is associated with aGVHD and can predict survival following HSCT [110,111,112,113,114,115,116,117,118,119]. However, gut microbiota diversity is often disrupted and declines during HSCT [113]. FMT, a therapeutic procedure involving the transfer of intestinal microbiota from healthy donors to recipients to restore a functionally competent microbiome, has emerged as a promising strategy for HSCT [120,121]. Recent studies published in PubMed regarding the therapeutic implications of FMT in HSCT are summarized in Table 3.

Several studies have indicated that FMT improves HSCT outcomes. Fujimoto et al., using gnotobiotic and humanized mice, demonstrated that purified endolysin suppressed aGVHD by reducing abundance of *Enterococcus faecalis* (and its cylLL gene), lowering serum IFN-γ levels, and improving survival without consistent alterations in microbial community structure, suggesting its potential clinical value for aGVHD management [122]. A randomized, double-blind, placebo-controlled phase II trial reported that allogeneic HSCT recipients and AML patients undergoing induction chemotherapy who underwent an FMT had lower infection densities at four months compared to patients given a placebo [123]. Similarly, van Lier et al. conducted a prospective, single-center, single-arm study involving 15 allo-HSCT recipients with steroid-resistant/dependent intestinal GVHD, revealing that unrelated-donor FMT was well-tolerated, achieved complete response in 10 patients (6 of whom could taper immunosuppressants), improved gut microbiome diversity, and increased *butyrate-producing bacteria*, with durable remission linked to improved 24-week survival [124]. Reddi et al. designed a randomized, double-blind, placebo-controlled trial in which they transplanted fecal microbiota from three donors into 20 patients, finding that FMT safely and effectively restored microbiota. Engraftment (associated with donor-like microbiota shifts and better clinical outcomes) was highest with Donor 3 (66%, significantly higher than the figures for Donors 1 and 2; *p* = 0.02 and *p* = 0.03, respectively) [125]. Several other reports have also corroborated these findings [126,127,128,129,130,131]. Malard et al. further demonstrated that FMT appeared to be safe and effective for restoring gut microbiota richness and diversity in AML patients receiving intensive chemotherapy and antibiotics [132].

Conversely, several small-scale studies have found potential adverse effects of FMT on GVHD remission in HSCT recipients. A randomized placebo-controlled trial involving 100 AML patients undergoing induction chemotherapy and HSCT recipients found that third-party FMT performed at the peak of a microbiota injury improved microbiome diversity, increased commensal bacteria quantities, and reduced quantities of potential pathogens but was associated with a higher frequency of acute GVHD compared to trials involving placebos [133]. Similarly, a review by Lo Porto et al. [134] concluded that despite promising results for managing recurrences in other patient populations, FMT is not routinely recommended in HSCT due to limited safety data. A randomized clinical trial by Ladas et al. [135] showed that although administration of *Lactobacillus plantarum* 299v was safe among children and adolescents undergoing allo-HSCT, it was ineffective in preventing gastrointestinal acute GVHD. Additionally, DeFilipp et al. [136] reported cases of bacteremia associated with FMT.

Although FMT is a promising therapeutic option for addressing microbiota disruption during or after cancer therapy, the existing research is limited by small sample sizes and short follow-up durations. To achieve clinical translation, rigorous risk–benefit analyses are necessary, and long-term outcomes and personalized treatment regimens require further investigation.

**Table 3 biomedicines-14-00055-t003:** FMT in hematopoietic stem cell transplantation.

Study	Subject	Disease	Intervention	Result (+/−)
Fujimoto et al. [122]	Gnotobiotic C57BL/6 mice mono-colonized with *E. faecalis*	Allo-HSCT	Purified endolysin against *E. faecalis*	Researchers noted potential clinical utility for reducing aGVHD risk.
Rashidi et al. [133]	100 patients	AML induction chemotherapy and allo-HSCT	FMT capsule (≥1 × 10^11^ bacteria, ≥40% viability)	FMT was associated with a higher risk for aGVHD in allo-HSCT recipients.
Mullish et al. [137]	50 adult patients	Allo-HSCT	Capsulized IMT performed shortly before HSCT conditioning	Results were incomplete.
Rashidi et al. [123]	74 patients	Allo-HSCT and AML	Third-party FMT (five oral capsules administered simultaneously)	FMT was safe and improved intestinal dysbiosis, but it did not reduce infection rates.
van Lier et al. [124]	15 patients	Allo-HSCT (AML, myelodysplastic syndrome, lymphoma, myeloproliferative disorder)	FMT from healthy volunteers with Western diets	Unrelated-donor FMT improved microbiome diversity, promoted activity of butyrate-producing bacteria, and achieved remission in steroid-resistant/dependent GvHD.
Reddi et al. [125]	20 patients	Allo-HSCT	FMT performed at a median of 25 days post-HSCT	FMT was safe, restored microbiota diversity, and increased quantities of beneficial commensal species.
Goeser, et al. [138]	11 Patients	Steroid-refractory GVHD	FMT via capsules	Treatment significantly reduced severity and staging of steroid-refractory GVHD.
Zhao et al. [127]	55 patients	Steroid-refractory acute gut GVHD	FMT after diagnosis of steroid-refractory GVHD	FMT significantly improved clinical remission and 90-day overall survival rates compared to controls.
Innes et al. [128]	19 patients	MDRO colonized HSCT	FMT administered 2–6 weeks before HSCT conditioning	Non-FMT patients had significantly lower survival rates and higher non-relapse mortality.
Sofi et al. [129]	C57BL/6 (B6; H-2^b^, CD45.2), B6.Ly5.1 (CD45.1), BD2F1 (H-2^b/d^), and BALB/c (H-2^d^) mice	Murine GVHD model	Dirty bedding/feces transferred pre- and post-BMT	Single-strain *Bacteroides fragilis* protected gut integrity and reduced GVHD.
Ladas et al. [135]	161 patients	Allo- HSCT	Administration of *L. plantarum* 299v from conditioning start to day 56 post-HSCT	Probiotics were ineffective in preventing gastrointestinal aGVHD.
Youngster et al. [130]	21 patients	Steroid-resistant/dependent gastrointestinal aGvHD	FMT via capsules	First FMT resulted in a 52.4% clinical response at 28 days, with increased abundance of beneficial Clostridiales and reduced abundance of pathogenic Enterobacteriales.
Yang et al. [139]	12 patients	Refractory chronic GVHD	FMT	The researchers observed increased gut microbial diversity, abundance of SCFA-producing bacteria, fecal SCFA levels, and levels of peripheral CD4(+)CD127(−) Treg cells; reduced pathogenic bacteria abundance and inflammatory cell infiltration; and enhanced colonic Treg infiltration.
Merli et al. [140]	5 pediatric patients	Allo-HSCT	FMT using samples from the same donor	Treatment led to 80% (4/5) MDR decolonization within one week post-FMT.
Rashidi et al. [131]	74 patients	T-cell replete allo-HSCT or AML chemotherapy	FMT capsule (≥1 × 10^11^ bacteria, ≥40% viability)	FMT potentially protects against aGVHD, especially in patients with severe microbiota disruptions.

allo-HSCT, allogenic hematopoietic cell transplantation; AML, acute myeloid leukemia; IMT, intestinal microbiota transplant; GVHD, graft-vs.-host disease; MDRO, multidrug-resistant organism; BMT, bone marrow transplantation; aGVHD, acute graft-versus-host disease; SCFA, short-chain fatty acid; MDR, multidrug resistant.

## 6. Conclusions

Neutrophils respond rapidly to threats such as infection and inflammation and play a pivotal role in defending against bacterial and fungal pathogens. Chemotherapy can significantly reduce neutrophil counts and even cause prolonged DNR, leading to a heightened risk of infection, which not only delays cancer treatment but may also reduce overall survival. Figure 1 showed the overview of this review.Extensive evidence suggests that the gut microbiota represents a promising predictive biomarker and therapeutic target for CIN. Despite encouraging findings, several key challenges must be addressed before gut microbiota can be widely applied in personalized medicine. Firstly, oncologists remain cautious regarding the clinical application of probiotics due to the potential risk of infection. In contrast, FMT appears to be relatively safe and may improve outcomes in HSCT; however, the current evidence base is limited by small sample sizes, selection bias, and inconsistent findings across studies. Therefore, researchers should prioritize large-scale, rigorously designed clinical trials. Secondly, the accuracy and reproducibility of microbiological data remain controversial, as results may vary due to host-related factors such as race, geography, environment, diet, and comorbidities, as well as differences in sampling protocols and sequencing methodologies [62]. These confounding factors should be carefully considered in future analyses to ensure reliable conclusions are obtained. Finally, while most studies have focused on reduced SCFA levels caused by gut microbiota dysbiosis, which may suppress neutrophil production, the underlying signaling pathways through which SCFAs regulate granulopoiesis remain largely unexplored.

In conclusion, integrating the gut microbiota insights into individualized clinical treatments is essential. The identification of specific gut microbes and their derived metabolites or enzymes that predict and modulate CIN could substantially improve cancer patients’ quality of life and prognoses. However, extensive research is still required before these findings can be translated into clinical practice. This study provides a theoretical foundation for developing “gut-protective” chemotherapy regimens, is of significant clinical relevance for improving cancer outcomes, and may open new avenues for the personalized application of chemotherapy.

## Data Availability

No new data were created or analyzed in this study.

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
