# Peer review of "Advances in Understanding the Impact of Human Gut Microbiota on Chemotherapy-Induced Neutropenia"

_biomedicines, 2025, doi:10.3390/biomedicines14010055_

Round 1

Reviewer 1 Report

Comments and Suggestions for Authors

Biomedicines-4008351

Type of manuscript: Review
Title: Advances in Understanding the Impact of Human Gut Microbiota on Chemotherapy-induced Neutropenia
Authors: Mengyuan He, Liangkang Lin, Su Liu *, Chun Chen *

This paper reviews the impact of chemotherapy-induced neutropenia on the human gut microbiota in cancer patients. It addresses chemotherapy-induced neutropenia (CIN), a major side effect of chemotherapy in many cancer patients. However, the paper contains numerous shortcomings, including the omission of the key keyword "CIN." Additional corrections are needed to address the issues noted below.

[Major concerns]

  1. chemotherapy-induced neutropenia (CIN): As mentioned above, define the term CIN specifically in the Introduction and then write your paper using the CIN abbreviation.
  2. References: I don't know when this review was written, but it doesn't cite any recent CIN-related review articles. When writing a review, you should cite other authors' review articles, mention their value, and then specifically describe how your review differs from these other articles.
  • The role of the gut microbiota in chemotherapy response, efficacy and toxicity: a systematic review. Böhm D, et al. NPJ Precis Oncol. 2025. PMID: 40738965.  
  • Gut diversity and the resistome as biomarkers of febrile neutropenia outcome in paediatric oncoloy patients undergoing hematopoietic stem cell transplantation. Sara Sardzikova1, Kristina Andrijkova 1, Peter Svec 2, Gabor Beke 3, Lubos Klucar 3, Gabriel Minarik 4, Viktor Bielik 5, Alexandra Kolenova 2, Katarina Soltys 6 Sci Rep. 2024 Mar 6;14(1):5504. doi: 10.1038/s41598-024-56242
  • The role of gut microbiota and metabolites in cancer chemotherapy. Li S, Zhu S, Yu J.J Adv Res. 2024 Oct;64:223-235. doi: 10.1016/j.jare.2023.11.027. Epub 2023 Nov 26.
  • A Literature Review on the Impact of the Gut Microbiome on Cancer Treatment Efficacy, Disease Evolution and Toxicity: The Implications for Hematological Malignancies. Ioana Gabriela Dumitru1, Samuel Bogdan Todor 1, Cristian Ichim 1, Claudiu Helgiu 1, Alina Helgiu 1 Clin Med. 2025 Apr 25;14(9):2982.  doi: 10.3390/jcm14092982.
  • A Review of Gut Microbiota-Derived Metabolites in Tumor Progression and Cancer Therapy. Yang Q, Wang B, Zheng Q, Li H, Meng X, Zhou F, Zhang L.Adv Sci (Weinh). 2023 May;10(15): e2207366. doi: 10.1002/advs.202207366. Epub 2023 Mar 23.
  • Cause and effect? A review of the impact of antibiotics on the gut microbiome in patients undergoing hematopoietic stem cell transplantation. Nakagaki M, et al. Expert Rev Anti Infect Ther. 2025. PMID: 40591953 Review.
  • Gut microbiota as a hidden modulator of chemotherapy: implications for colorectal cancer treatment. Sadeghloo Z, et al. Discov Oncol. 2025. PMID: 41003873 
  1. The nomenclature of gut microbes: The general rule is to italicize their genus and species names, but this was not followed at all.
  2. Describe in more detail the gastrointestinal complications (diarrhea, mucositis) associated with clinical consequences of CIN-associated microbiome dysregulation.
  3. There are numerous anticancer drugs used in cancer treatment, and a section on "chemotherapeutic agents and their microbiome signatures" is needed. Specifically, describe the most frequently used anticancer drugs, including alkylating agents (e.g., cyclophosphamide), antimetabolites (5-FU, methotrexate), anthracyclines, platinum compounds, taxanes, and known patterns of microbiome alteration for different drug classes.
  4. Microbiome-based biomarkers in CIN patients, especially those related to 1) microbial signatures predicting risk of febrile neutropenia; 2) resistome analysis and infection risk; and 3) microbiome features associated with chemotherapy response, should be added.
  5. Abbreviations: Abbreviations are useful not only for keeping terminology concise but also for ensuring clarity and consistency within a manuscript. When first introducing an abbreviated term, write out the full expression followed by the abbreviation in parentheses. After this initial definition, use the abbreviation uniformly throughout all sections. Reserve abbreviations for terms that occur multiple times; if a term appears only once, it is preferable to spell it out in full.
  6. Abbreviations in Tables and Figures: When abbreviations appear in tables or figures, provide their full forms either in the accompanying legends or directly below the tables. This practice helps maintain transparency and improves readability for the audience. In this paper, abbreviations and full names are written fairly well, but if it is not a proper noun, do not capitalize the full name of the abbreviation, but write it in lowercase. In this paper, abbreviations and full names are written fairly well, but please correct them as in the following example. Example at Table 1: aGVHD, acute graft-versus-host disease; ALL, acute lymphoblastic leukemia; allo-HSCT, allogeneic hematopoietic stem 331 cell transplantation; CID, chemotherapy-induced diarrhea; NP, no report; RCT, randomised controlled trial.

[Minor concerns]

  1. Line 27: Update the cancer incidence statistics paper to the latest paper.
  2. Line 63: Define HR and CI.
  3. Line 91: Define PFS and OS.
  4. Line 136: Define Although HSCT is defined in the tables, it should be defined even if it is used for the first time in the text.
  5. Line 193: Define BCoAT and explain it here.
  6. Line 197 N-acetylglucosamine: 'N' should be italicized.
  7. Line 251: Define 5-FU.
  8. Table 1 ‘Tumor’ vs.’ Tumour’: American English uses "tumor" and British English uses "tumour." In a given paper, only one of these two terms should be used.
  9. When listing the main authors in Tables 1 and 2, list only the last name.
  10. Line 338: Rewrite it.
  11. Figure 1: Check again the names of the cells shown in Figure 1. They are probably ‘Neutrophil’, but they are written in very poor English.

Overall, the manuscript should go through a major revision as indicated above.

Comments on the Quality of English Language

Biomedicines-4008351

Type of manuscript: Review
Title: Advances in Understanding the Impact of Human Gut Microbiota on Chemotherapy-induced Neutropenia
Authors: Mengyuan He, Liangkang Lin, Su Liu *, Chun Chen *

This paper reviews the impact of chemotherapy-induced neutropenia on the human gut microbiota in cancer patients. It addresses chemotherapy-induced neutropenia (CIN), a major side effect of chemotherapy in many cancer patients. However, the paper contains numerous shortcomings, including the omission of the key keyword "CIN." Additional corrections are needed to address the issues noted below

[Major concerns]

  1. chemotherapy-induced neutropenia (CIN): As mentioned above, define the term CIN specifically in the Introduction and then write your paper using the CIN abbreviation.
  2. References: I don't know when this review was written, but it doesn't cite any recent CIN-related review articles. When writing a review, you should cite other authors' review articles, mention their value, and then specifically describe how your review differs from these other articles.
  • The role of the gut microbiota in chemotherapy response, efficacy and toxicity: a systematic review. Böhm D, et al. NPJ Precis Oncol. 2025. PMID: 40738965.  
  • Gut diversity and the resistome as biomarkers of febrile neutropenia outcome in paediatric oncoloy patients undergoing hematopoietic stem cell transplantation. Sara Sardzikova1, Kristina Andrijkova 1, Peter Svec 2, Gabor Beke 3, Lubos Klucar 3, Gabriel Minarik 4, Viktor Bielik 5, Alexandra Kolenova 2, Katarina Soltys 6 Sci Rep. 2024 Mar 6;14(1):5504. doi: 10.1038/s41598-024-56242
  • The role of gut microbiota and metabolites in cancer chemotherapy. Li S, Zhu S, Yu J.J Adv Res. 2024 Oct;64:223-235. doi: 10.1016/j.jare.2023.11.027. Epub 2023 Nov 26.
  • A Literature Review on the Impact of the Gut Microbiome on Cancer Treatment Efficacy, Disease Evolution and Toxicity: The Implications for Hematological Malignancies. Ioana Gabriela Dumitru1, Samuel Bogdan Todor 1, Cristian Ichim 1, Claudiu Helgiu 1, Alina Helgiu 1 Clin Med. 2025 Apr 25;14(9):2982.  doi: 10.3390/jcm14092982.
  • A Review of Gut Microbiota-Derived Metabolites in Tumor Progression and Cancer Therapy. Yang Q, Wang B, Zheng Q, Li H, Meng X, Zhou F, Zhang L.Adv Sci (Weinh). 2023 May;10(15): e2207366. doi: 10.1002/advs.202207366. Epub 2023 Mar 23.
  • Cause and effect? A review of the impact of antibiotics on the gut microbiome in patients undergoing hematopoietic stem cell transplantation. Nakagaki M, et al. Expert Rev Anti Infect Ther. 2025. PMID: 40591953 Review.
  • Gut microbiota as a hidden modulator of chemotherapy: implications for colorectal cancer treatment. Sadeghloo Z, et al. Discov Oncol. 2025. PMID: 41003873 
  1. The nomenclature of gut microbes: The general rule is to italicize their genus and species names, but this was not followed at all.
  2. Describe in more detail the gastrointestinal complications (diarrhea, mucositis) associated with clinical consequences of CIN-associated microbiome dysregulation.
  3. There are numerous anticancer drugs used in cancer treatment, and a section on "chemotherapeutic agents and their microbiome signatures" is needed. Specifically, describe the most frequently used anticancer drugs, including alkylating agents (e.g., cyclophosphamide), antimetabolites (5-FU, methotrexate), anthracyclines, platinum compounds, taxanes, and known patterns of microbiome alteration for different drug classes.
  4. Microbiome-based biomarkers in CIN patients, especially those related to 1) microbial signatures predicting risk of febrile neutropenia; 2) resistome analysis and infection risk; and 3) microbiome features associated with chemotherapy response, should be added.
  5. Abbreviations: Abbreviations are useful not only for keeping terminology concise but also for ensuring clarity and consistency within a manuscript. When first introducing an abbreviated term, write out the full expression followed by the abbreviation in parentheses. After this initial definition, use the abbreviation uniformly throughout all sections. Reserve abbreviations for terms that occur multiple times; if a term appears only once, it is preferable to spell it out in full.
  6. Abbreviations in Tables and Figures: When abbreviations appear in tables or figures, provide their full forms either in the accompanying legends or directly below the tables. This practice helps maintain transparency and improves readability for the audience. In this paper, abbreviations and full names are written fairly well, but if it is not a proper noun, do not capitalize the full name of the abbreviation, but write it in lowercase. In this paper, abbreviations and full names are written fairly well, but please correct them as in the following example. Example at Table 1: aGVHD, acute graft-versus-host disease; ALL, acute lymphoblastic leukemia; allo-HSCT, allogeneic hematopoietic stem 331 cell transplantation; CID, chemotherapy-induced diarrhea; NP, no report; RCT, randomised controlled trial.

[Minor concerns]

  1. Line 27: Update the cancer incidence statistics paper to the latest paper.
  2. Line 63: Define HR and CI.
  3. Line 91: Define PFS and OS.
  4. Line 136: Define Although HSCT is defined in the tables, it should be defined even if it is used for the first time in the text.
  5. Line 193: Define BCoAT and explain it here.
  6. Line 197 N-acetylglucosamine: 'N' should be italicized.
  7. Line 251: Define 5-FU.
  8. Table 1 ‘Tumor’ vs.’ Tumour’: American English uses "tumor" and British English uses "tumour." In a given paper, only one of these two terms should be used.
  9. When listing the main authors in Tables 1 and 2, list only the last name.
  10. Line 338: Rewrite it.
  11. Figure 1: Check again the names of the cells shown in Figure 1. They are probably ‘Neutrophil’, but they are written in very poor English.

Overall, the manuscript should go through a major revision as indicated above.

Author Response

[Major concerns]

  1. chemotherapy-induced neutropenia (CIN): As mentioned above, define the term CIN specifically in the Introduction and then write your paper using the CIN abbreviation.

Revisions are completed, and the modified content is highlighted in red.

2.References: I don't know when this review was written, but it doesn't cite any recent CIN-related review articles. When writing a review, you should cite other authors' review articles, mention their value, and then specifically describe how your review differs from these other articles.

  • The role of the gut microbiota in chemotherapy response, efficacy and toxicity: a systematic review. Böhm D, et al. NPJ Precis Oncol. 2025. PMID: 40738965.  
  • Gut diversity and the resistome as biomarkers of febrile neutropenia outcome in paediatric oncoloy patients undergoing hematopoietic stem cell transplantation. Sara Sardzikova1, Kristina Andrijkova1, Peter Svec 2, Gabor Beke 3, Lubos Klucar 3, Gabriel Minarik 4, Viktor Bielik 5, Alexandra Kolenova 2, Katarina Soltys 6 Sci Rep. 2024 Mar 6;14(1):5504. doi: 10.1038/s41598-024-56242
  • The role of gut microbiota and metabolites in cancer chemotherapy. Li S, Zhu S, Yu J.J Adv Res. 2024 Oct;64:223-235. doi: 10.1016/j.jare.2023.11.027. Epub 2023 Nov 26.
  • A Literature Review on the Impact of the Gut Microbiome on Cancer Treatment Efficacy, Disease Evolution and Toxicity: The Implications for Hematological Malignancies. Ioana Gabriela Dumitru1, Samuel Bogdan Todor1, Cristian Ichim 1, Claudiu Helgiu 1, Alina Helgiu 1 Clin Med. 2025 Apr 25;14(9):2982.  doi: 10.3390/jcm14092982.
  • A Review of Gut Microbiota-Derived Metabolites in Tumor Progression and Cancer Therapy. Yang Q, Wang B, Zheng Q, Li H, Meng X, Zhou F, Zhang L.Adv Sci (Weinh). 2023 May;10(15): e2207366. doi: 10.1002/advs.202207366. Epub 2023 Mar 23.
  • Cause and effect? A review of the impact of antibiotics on the gut microbiome in patients undergoing hematopoietic stem cell transplantation. Nakagaki M, et al. Expert Rev Anti Infect Ther. 2025. PMID: 40591953 Review.
  • Gut microbiota as a hidden modulator of chemotherapy: implications for colorectal cancer treatment. Sadeghloo Z, et al. Discov Oncol. 2025. PMID: 41003873 

Revisions are completed, and the modified content is highlighted in red

The nomenclature of gut microbes: The general rule is to italicize their genus and species names, but this was not followed at all.

Revisions are completed, and the modified content is highlighted in red

Describe in more detail the gastrointestinal complications (diarrhea, mucositis) associated with clinical consequences of CIN-associated microbiome dysregulation.

Revisions are completed, and the modified content is highlighted in red

There are numerous anticancer drugs used in cancer treatment, and a section on "chemotherapeutic agents and their microbiome signatures" is needed. Specifically, describe the most frequently used anticancer drugs, including alkylating agents (e.g., cyclophosphamide), antimetabolites (5-FU, methotrexate), anthracyclines, platinum compounds, taxanes, and known patterns of microbiome alteration for different drug classes.

Revisions are completed, and the modified content is highlighted in red

Microbiome-based biomarkers in CIN patients, especially those related to 1) microbial signatures predicting risk of febrile neutropenia; 2) resistome analysis and infection risk; and 3) microbiome features associated with chemotherapy response, should be added.

Revisions are completed, and the modified content is highlighted in red

Abbreviations: Abbreviations are useful not only for keeping terminology concise but also for ensuring clarity and consistency within a manuscript. When first introducing an abbreviated term, write out the full expression followed by the abbreviation in parentheses. After this initial definition, use the abbreviation uniformly throughout all sections. Reserve abbreviations for terms that occur multiple times; if a term appears only once, it is preferable to spell it out in full.

Revisions are completed, and the modified content is highlighted in red

Abbreviations in Tables and Figures: When abbreviations appear in tables or figures, provide their full forms either in the accompanying legends or directly below the tables. This practice helps maintain transparency and improves readability for the audience. In this paper, abbreviations and full names are written fairly well, but if it is not a proper noun, do not capitalize the full name of the abbreviation, but write it in lowercase. In this paper, abbreviations and full names are written fairly well, but please correct them as in the following example. Example at Table 1: aGVHD, acute graft-versus-host disease; ALL, acute lymphoblastic leukemia; allo-HSCT, allogeneic hematopoietic stem 331 cell transplantation; CID, chemotherapy-induced diarrhea; NP, no report; RCT, randomised controlled trial.

Revisions are completed, and the modified content is highlighted in red

[Minor concerns]

  1. Line 27: Update the cancer incidence statistics paper to the latest paper.

Revisions are completed, and the modified content is highlighted in red

  1. Line 63: Define HR and CI.

Revisions are completed, and the modified content is highlighted in red

  1. Line 91: Define PFS and OS.

Revisions are completed, and the modified content is highlighted in red

  1. Line 136: Define Although HSCT is defined in the tables, it should be defined even if it is used for the first time in the text.

Revisions are completed, and the modified content is highlighted in red

  1. Line 193: Define BCoAT and explain it here.

Revisions are completed, and the modified content is highlighted in red

  1. Line 197 N-acetylglucosamine: 'N' should be italicized.

Revisions are completed, and the modified content is highlighted in red

  1. Line 251: Define 5-FU.

It has already been defined when it first appeared in the preceding part.

  1. Table 1 ‘Tumor’ vs.’ Tumour’: American English uses "tumor" and British English uses "tumour." In a given paper, only one of these two terms should be used.

We use “Tumor” in this paper.

  1. When listing the main authors in Tables 1 and 2, list only the last name.

We have already made the correction

  1. Line 338: Rewrite it.

We have already made the correction

  1. Figure 1: Check again the names of the cells shown in Figure 1. They are probably ‘Neutrophil’, but they are written in very poor English.

We have already made the correction

Reviewer 2 Report

Comments and Suggestions for Authors

This review article describes the literature data regarding the communication of the innate immune system and microbiota, a complex process that the authors named "gut-protective" chemotherapy.  They update on various clinical studies that investigated the depletion of neutrophils along treatment regimens with chemotherapy agents to cancer patients. In the sections, the authors present and discuss molecular mechanisms and adverse effects of chemotherapies to neutrophils and granulopoiesis and consequences, mainly elevated risk of infections, to cancer patients. The representative cases are shown in detail in Table 1. In the second part of the text, the authors present and discuss some current mechanisms, focusing on dysbiosis and metabolites. Finally, the authors discuss fecal microbiota transplantation, prebiotics, probiotics, synbiotics, and other therapeutics. The clinical trial data are presented in Table 2 and 3. In conclusion, the authors state "to achieve clinical translation, rigorous risk-benefit analyses are necessary, ...". Overall the manuscript addressed many important issues regarding the clinical evidence and ongoing research in the area of microbiome and the immune system. However, the authors missed a critical point on treatments for chemotherapy-induced neutropenia, which is a post-treatment using some labels of the growth factors like G-CSFs. In the revised version of the manuscript the authors should specifically discuss this issue and its impact on dysbiosis and patient’s outcomes. Finally, ensure the text is free from grammatical errors, spelling, and italicized name of bacterial species, soon on. 

Comments on the Quality of English Language

No comments

Author Response

Please see the attachment。

Round 2

Reviewer 1 Report

Comments and Suggestions for Authors

Biomedicines-4008351-v2

Type of manuscript: Review

Title: Advances in Understanding the Impact of Human Gut Microbiota on Chemotherapy-induced Neutropenia

Authors: Mengyuan He, Liangkang Lin, Su Liu *, Chun Chen *

The authors stated that they had made revisions to the issues raised during the first review, and the quality of the paper has significantly improved thanks to the comments of other reviewers. However, the English writing in the newly corrected sentences in red is lower than that of the first submission. It seems that the second submission did not receive proper English proofreading. I hope that The Office will take responsibility for correcting this during the proofreading process.

[Minor concerns]

  1. Parentheses in the middle of English sentences should be separated from the preceding word.
  2. Lines 43, 57, 255, etc.: When citing the first author's name in the text, only the last name should be indicated, but these three examples include the first name's initial. Please correct similar typos.
  3. In addition to those pointed out above, there are many minor typos, so please correct them.

Overall, accept the current manuscript after minor revision.

Comments on the Quality of English Language

Biomedicines-4008351-v2

Type of manuscript: Review

Title: Advances in Understanding the Impact of Human Gut Microbiota on Chemotherapy-induced Neutropenia

Authors: Mengyuan He, Liangkang Lin, Su Liu *, Chun Chen *

The authors stated that they had made revisions to the issues raised during the first review, and the quality of the paper has significantly improved thanks to the comments of other reviewers. However, the English writing in the newly corrected sentences in red is lower than that of the first submission. It seems that the second submission did not receive proper English proofreading. I hope that The Office will take responsibility for correcting this during the proofreading process.

[Minor concerns]

  1. Parentheses in the middle of English sentences should be separated from the preceding word.
  2. Lines 43, 57, 255, etc.: When citing the first author's name in the text, only the last name should be indicated, but these three examples include the first name's initial. Please correct similar typos.
  3. In addition to those pointed out above, there are many minor typos, so please correct them.

Overall, accept the current manuscript after minor revision.

Author Response

  1. Parentheses in the middle of English sentences should be separated from the preceding word.

    We have already made the correction

  2. Lines 43, 57, 255, etc.: When citing the first author's name in the text, only the last name should be indicated, but these three examples include the first name's initial. Please correct similar typos.

    Revisions are completed, and the modified content is highlighted in purple.

  3. In addition to those pointed out above, there are many minor typos, so please correct them.Revisions are completed, and the modified content is highlighted in purple.